# Preclinical Evaluation of a Novel Small Molecule LCC-21 to Suppress Colorectal Cancer Malignancy by Inhibiting Angiogenic and Metastatic Signatures

**DOI:** 10.3390/cells12020266

**Published:** 2023-01-09

**Authors:** Ntlotlang Mokgautsi, Yu-Cheng Kuo, Yan-Jiun Huang, Chien-Hsin Chen, Debabrata Mukhopadhyay, Alexander T. H. Wu, Hsu-Shan Huang

**Affiliations:** 1Ph.D. Program for Cancer Biology and Drug Discovery, College of Medical Science and Technology, Taipei Medical University and Academia Sinica, Taipei 11031, Taiwan; 2Graduate Institute for Cancer Biology and Drug Discovery, College of Medical Science and Technology, Taipei Medical University, Taipei 11031, Taiwan; 3Department of Pharmacology, School of Medicine, College of Medicine, Taipei Medical University, Taipei 11031, Taiwan; 4School of Post-Baccalaureate Chinese Medicine, College of Chinese Medicine, China Medical University, Taichung 40402, Taiwan; 5Division of Colorectal Surgery, Department of Surgery, Taipei Medical University Hospital, Taipei Medical University, Taipei 11031, Taiwan; 6Department of Surgery, College of Medicine, Taipei Medical University, Taipei 11031, Taiwan; 7School of Medicine, College of Medicine, Taipei Medical University, Taipei 11031, Taiwan; 8Division of Colorectal Surgery, Department of Surgery, WanFang Hospital, Taipei Medical University, No. 111 Sec. 3 Xinglong Rd., Wenshan Dist., Taipei 11031, Taiwan; 9Department of Biochemistry and Molecular Biology, Mayo Clinic, Rochester, MN 55905, USA; 10TMU Research Center of Cancer Translational Medicine, Taipei Medical University, Taipei 11031, Taiwan; 11The Ph.D. Program of Translational Medicine, College of Medical Science and Technology, Taipei Medical University, Taipei 11031, Taiwan; 12Clinical Research Center, Taipei Medical University Hospital, Taipei Medical University, Taipei 11031, Taiwan; 13Graduate Institute of Medical Sciences, National Defense Medical Center, Taipei 11031, Taiwan; 14School of Pharmacy, National Defense Medical Center, Taipei 11031, Taiwan; 15Ph.D. Program in Biotechnology Research and Development, College of Pharmacy, Taipei Medical University, Taipei 11031, Taiwan

**Keywords:** colorectal cancer (CRC), LCC-21, niclosamide, angiogenesis, metastasis, stemness, therapeutic resistance

## Abstract

Colorectal cancer (CRC) is one of the most common cancers, and it frequently metastasizes to the liver and lymph nodes. Despite major advances in treatment modalities, CRC remains a poorly characterized biological malignancy, with high reported cases of deaths globally. Moreover, cancer stem cells (CSCs) and their microenvironment have been widely shown to promote colon cancer development, progression, and metastasis. Therefore, an understanding of the underlying mechanisms that contribute to the maintenance of CSCs and their markers in CRC is crucial in efforts to treat cancer metastasis and develop specific therapeutic targets for augmenting current standard treatments. Herein, we applied computational simulations using bioinformatics to identify potential theranostic markers for CRC. We identified the overexpression of vascular endothelial growth factor-α (*VEGFA*)/β-catenin/matrix metalloproteinase (MMP)-7/Cluster of Differentiation 44 (*CD44*) in CRC to be associated with cancer progression, stemness, resistance to therapy, metastasis, and poor clinical outcomes. To further investigate, we explored in silico molecular docking, which revealed potential inhibitory activities of LCC-21 as a potential multitarget small molecule for *VEGF-A/CTNNB1/MMP7/CD44* oncogenic signatures, with the highest binding affinities displayed. We validated these finding in vitro and demonstrated that LCC-21 inhibited colony and sphere formation, migration, and invasion, and these results were further confirmed by a Western blot analysis in HCT116 and DLD-1 cells. Thus, the inhibitory effects of LCC-21 on these angiogenic and onco-immunogenic signatures could be of translational relevance as potential CRC biomarkers for early diagnosis.

## 1. Introduction

Colorectal cancer (CRC) is the second most common cancer, with continually reported cases and high mortality globally [1,2]. CRC is a heterogeneous disease with varying molecular characteristics and clinical outcomes in patients; consequently, the cancer is diagnosed in an advanced stage, and approximately 25% of patients present with localized or distant metastasis [3,4,5,6]. Amongst the most common clinical phenotypes of CRC are cancer stemness angiogenesis, resistance to chemotherapy, and metastasis [7,8]. Despite advanced molecularly stratified therapeutic interventions, including resection surgery, radiation, chemotherapeutics, and targeted therapeutics, the 5-year overall survival rate is still limited in CRC and is only 12% for metastatic (mCRC) patients [9,10,11]. In addition, submucosal invasion was reported to metastasize to lymph nodes in more than 10% of CRC cases [12]. Moreover, several studies have shown that CRC patients’ deaths were associated with liver metastasis; however, the mechanisms leading to cancer metastasis still remain to be fully explored [13,14,15,16].

CRC patients presently have far more treatment options and can produce remarkable responses. Chemotherapy is still commonly used as the mainstream approach of treatment [17,18]. 5-Fluorouracil (5-FU) was the first chemotherapy to exhibit satisfactory activities against CRC. For patients with mCRC, a combination of 5-FU and oxaliplatin has been used as first-line treatment; however, most patients become resistant to this treatment, indicating the need for new therapies that can produce dramatic improvements [19,20,21,22]. In addition to these combined chemotherapeutics for mCRC, targeted agents including anti-vascular endothelial growth factor-α (*VEGFA*) agents (such as bevacizumab and sorafenib) are used [23,24,25]. Angiogenesis is known to play an important role in the development of tumor growth and metastasis [26]. Recently, *VEGF*, a potent cytokine, was explored as an angiogenesis factor, commonly linked to CRC distant metastasis [16,27,28]. *VEGF* acts by binding to the tyrosine kinase receptor, *VEGF* receptor (*VEGFR*), which is expressed by the vascular endothelium [29,30].

Moreover, the activation of the signal transducer and activator of transcription-3 (*STAT3*), a member of the *STAT* family of transcription factors, which modulates various biological processes and is an attractive treatment target in CRC [31], was shown to upregulate *VEGF-A* in primary and metastatic CRC, and signals a poor prognosis [32,33,34,35,36]. The Wnt/β-catenin signaling pathway plays a significant role in CRC tumorigenesis, stemness, progression, and metastasis [37,38,39]. Accumulated studies have demonstrated that the activation of β-catenin *(CTNNB1)* is associated with angiogenesis and cancer metastasis by modulating the expression of *VEGF* and the matrix metalloproteinases (*MMPs*) pathway in CRC [40,41]. *MMPs* are extracellular proteases that target cytokines and receptors [42,43]; moreover, one member of the *MMP* family (*MMP7*) is widely overexpressed in approximately 80% of CRC cases, and is associated with tumor neovascularization, invasion, and distant lymph node metastasis [44,45,46,47,48,49]. Moreover, accumulating reports have shown that *MMP7* regulates *VEGF* pathways via the degradation of *VEGFR1*, which subsequently promotes angiogenesis [50,51]. Inhibition of angiogenesis is considered to be an effective approach to suppress tumor progression [52]. Furthermore, colorectal cancer stem cells (CSCs) have been demonstrated to alter the tumor microenvironment (TME) by regulating expression angiogenesis factors, which ultimately increase neovascularization outside the site of metastasis [8,53,54]. Therefore, to analyze further, we explored single-cell RNA sequencing (scRNA-seq), which provides detailed characterization of tumor heterogeneity and tumor-associated immune cells in CRC [55,56]. Accordingly, we applied the publicly available multimodal omics scRNA-seq datasets and found a high abundance of *VEGFA/CTNNB1/MMP7/CD44* genes in CRC immune cells within the TME compared with the malignant cells and stromal cells. These results suggest that a link among *VEGFA/CTNNB1/MMP7/CD44* promotes cancer stemness, angiogenesis, metastasis, and poor prognoses in CRC.

Natural compounds have led to great advancements in the development of anticancer drugs, with potential to inhibit tumorigenesis and metastasis through targeting several signaling pathways, with low toxicity against normal cells [57,58,59,60,61]. Previously in our lab, a series of new 5-(2’,4’-difluorophenyl)-niclosamide derivatives were synthesized and evaluated for their anticancer activities. The development of 5-(2′,4′-difluorophenyl)-niclosamide derivatives based on difluorobiphenyl and niclosamide scaffolds were synthesized as previously described in a detailed protocol [62,63]. Two of these are NSC765689 (N-(4-cyanophenyl)-2’,4’-difluoro-4-hydroxy-[1,1’-biphenyl]-3-carboxamide), a closed ring structure, and its predecessor NSC828787 (N-(3-cyanophenyl)-2’,4’-difluoro-4-hydroxy-[1,1’-biphenyl]-3-carboxamide), an open ring structure, which consists of functional fragments of magnolol, 2,4-difluorophenyl, and niclosamide. In this study, we evaluated the anticancer activities of a novel small molecule, NSC765689 (LCC-21), as a target drug for *VEGFA/CTNNB1/MMP7/CD44* in CRC (Figure 1).

### Chemical Synthesis of the Novel Multi-Target Small Molecules NSC828787 (LCC09) and NSC765689 (LCC-21)

The title compounds were synthesized from 3-aminobenzonitrile as previously described in a detailed protocol [62,64]. Briefly, a solution of 0.95 g in 30 mL of anhydrous tetrahydrofuran (THF) was added dropwise to thionyl chloride 1 mL (SOCl₂). The mixed solution was refluxed under nitrogen atmosphere for eight hours, allowed to cool at room temperature, and immediately steamed. The residue was directly mixed with 0.5 mL of 4-chloro-2-fluoroaniline in 30 mL of anhydrous THF for 14 h. The THF was further removed, and the ethyl acetate was used to wash and extract the crude product. The product was further washed with 15 mL of 10% Sodium bicarbonate, 25 mL of water three times, and 10 mL brine, and it was then dried over anhydrous MgSO_4_. The crude product was purified by crystallization from hot EtOH and pure compounds were obtained as beige powder (yield 51%) for N-(3-cyanophenyl)-2’,4’-difluoro-4-hydroxy-[1,1’-biphenyl]-3-carboxamide (NSC828787) and yellowish powder (yield 54%) for N-(4-cyanophenyl)-2’,4’-difluoro-4-hydroxy-[1,1’-biphenyl]-3-carboxamide (NSC765689) [65] (Figure 2).

## 2. Materials and Methods

### 2.1. Microarray Dataset Extraction

A total of two (2) gene expression datasets (GSE81558 and GSE110223) were retrieved from the National Center for Biotechnology Information (NCBI), Gene Expression Omnibus (GEO; https://www.ncbi.nlm.nih.gov/geo/ 26 April 2022) [66,67]. GSE81558 contained 51 samples, including 19 CRC live metastasis samples and 23 primary CRC tumor and normal samples [68], and GSE110223 contained 26 samples, including 13 CRC cancer samples and 13 adjacent noncancerous samples [69]. The obtained datasets were further analyzed by GEO2R. The adjusted *p* value (adj.p) to control the false-discovery rate (FDR) and detection of possible false positives, as well as the identification of significant genes, was made using the Benjamin–Hochberg method. The fold-change (FC) threshold was set to 2 and adj. *p* < 0.05 was considered significant. Venn diagrams were constructed using the Bioinformatics & Evolutionary Genomics (BEG) online tool (http://bioinformatics.psb.ugent.be/webtools/Venn/ 22 May 2022).

### 2.2. Analysis of Pharmacokinetic (PK), Drug-Likeness, and Medicinal Chemical Properties of NSC765689 (LCC-21)

The pharmacokinetics (PK), drug-likeness, absorption, distribution, metabolism, and excretion (ADME) of NSC765689 were assessed using the SwissADME bioinformatics tool (http://www.swissadme.ch/ 23 May 2022) [70]. The criteria used for drug-likeness in drug discovery was evaluated based on the Lipinski (Pfizer) rule-of-five [71]. Furthermore, we predicted the blood–brain barrier (BBB) by utilizing the online (BBB) Prediction Server (https://www.cbligand.org/BBB/ 23 May 2022) [72]. To analyze further, we predicted the target genes for NSC76589 compound by using the Swiss target prediction tool (http://www.swisstargetprediction.ch/ 23 May 2022). The data used were collected from different public servers, including ChEMBL and PubChem, and all predictions were based on “probability” obtained from the target score to assess the plausibility of the predicted target being accurate [73].

### 2.3. Validation of Differentially Expressed Genes (DEGs) in Colorectal Cancer Cohorts

To validate the expression of the target oncogenes, we applied the Starbase online platform (https://starbase.sysu.edu.cn/ 26 May 2022) [74]. We further determined the survival probability ratio between CRC patients with low and high gene expressions using the DrugSurv online tool (http://www.bioprofiling.de/ 28 May 2022). For further analysis, we investigated the correlation among the expressed target genes using the STRING database (https://stringdb.org/ 28 May 2022), which allowed us to predict the protein–protein interactions (PPIs), and we further predicted the gene–gene interactions (GGI) using the Gene-mania tool (https://genemania.org/ 28 May 2022), under high confidence (with a minimal interaction score of 0.700) Furthermore, we explored the TMN plot (https://tnmplot.com/analysis/ 4 June 2022) to compare the expression of these oncogenes in tumor and metastatic CRC samples from RNA sequencing data (RNAseq), but using the Kruskal–Walls test to compare data, and finally identified the correlations among the target gene signatures using the cBioPortal (https://www.cbioportal.org/ 5 June 2022) online platform.

### 2.4. Functional Enrichment Analysis

Interactive networks obtained from STRING analysis were used to process the functional enrichment analysis, which included enriched biological processes and biological pathways. These enrichment analyses were performed using the DAVID bioinformatics webtool (https://david.ncifcrf.gov/tools.jsp 12 June 2022) [75], and further visualized by FunRich software (http://www.funrich.org/ 12 June 2022) [76]. To analyze further, we employed NetworkAnalyst, a comprehensive gene expression profiling and network visual analytics [77], using the SIGnaling Network Open Resource (SIGNOR 2.0) and selected the KEGG database to analyze enriched co-expressed genes from the platform (https://www.networkanalyst.ca/ 12 June 2022) [78], with *p* < 0.05 considered significant.

### 2.5. Analysis of Genomic Alterations of Targeted Genes and Immune infiltration in CRC

The genomic alterations of targeted *onco*genic signatures were analyzed using the Oncoprint feature of cBioportal software. Moreover, we utilized the tumor infiltrating immune cells tool (TIMER) (http://timer.cistrome.org/ 24 June 2022) [79] to analyze and visualize the effects of targeted gene mutations on immune cell infiltration in CRC and further applied TIMER 2.0 software (https://www.cistrome.shinyapps.io/timer 24 June 2022) [80] to identify the relationship between targeted gene expressions with selected immune cells; we applied a correlation analysis between these oncogenes and immune infiltration cells.

### 2.6. Bioinformatics Approaches to Single-Cell RNA-seq (scRNA-seq) Analysis

Single cell RNA sequencing of human CRC, HCC, was based on previously published scRNA-seq datasets [81]; herein, we explored publicly available scRNA-seq datasets and single-cell RNA sequencing (scRNA-seq) datasets of CRC (https://www.ebi.ac.uk/arrayexpress/ 26 June 2022), which included M-E-MAT-8410 and (9 patients; GSE144735) CRC tumors. We also matched adjacent normal colon tissue for the purpose of generating a cellular map of CRCs and their tumor microenvironment.

### 2.7. In Vitro Screening of LCC-21 against Full National Cancer Institute (NCI) Panels of Colorectal Tumor Cell Lines

LCC-21 was screened for anticancer activities on a panel of six (6) CRC tumor cell lines, including COLO205, HCC-2998, HCT-116, HCT-15, KM12, and SW620 from NCI, according to the outlined protocol of NCI (https://dtp.cancer.gov/ 28 June 2022) [82,83,84,85]. The compound was first evaluated for its antiproliferative and cytotoxic activities with an initial single dose of 10 μM, which was further administered, in a dose dependent manner, on the CRC cell line. We further explored the online expression atlas database tool, using the RNA-Seq mRNA baseline.

### 2.8. Receptor–Ligand Interaction Analysis

Molecular docking analysis was performed to determine the receptor–ligand interactions; briefly, we used ChemDraw Ultra 12.0 to construct the 3D structure of LCC-21 small molecules in mol2 format (https://ChemDraw Ultra 12.0/ 30 May 2022) [86]. The structure was later converted from mol2 into PDB formal using Pymol visualization software (https://pymol.org/2/ 30 May 2022) [87]. To analyze further, we utilized the protein databank website (https://www.rcsb.org/ 30 May 2022) to download the crystal structures of the target receptors: VEGFA (PDB:JO2B), CTNNB1 (PDB: 1JDH), MMP7 (2Y6C), and CD44 (PDB:1UUH). The structures were retrieved in PDB format. Th docking process requires PDBQT file format; accordingly, we converted all the PDB files, i.e.*,* receptors and ligand, to PDBQT file format using autodock tools, and proceeded to perform docking analysis [87].

### 2.9. Cell Culture and Reagents

Both DLD-1 and HCT116 human colon cancer cell lines were purchased from the American Type Culture Collection (ATCC, Manassas, VA, United States of America). Briefly, each cell line was cultured, then passaged at 90% cell confluency in Dulbecco*’*s Modified Eagle Medium (DMEM) (Invitrogen, Life Technologies, Carlsbad, CA, USA); cells were then stored under standard incubator conditions (in 5% humidified CO_2_ at 37 °C).

### 2.10. Cell Viability Assay

The cell viability assay was performed using sulforhodamine B (SRB) reagent (Sigma-Aldrich, Taipei, Taiwan), as described previously [88]. Briefly, DLD-1 and HCT116 cells were harvested at 90*–*95% confluency, and 8000 cells/well were seeded in 96-well plates for a period of 24 h, followed by dose-dependent concentration treatment of LCC-21. After 48 h of treatment, 10% trichloroacetic acid (TCA) was added to cells and they were stored at 4 °C for one hour. Cells were further washed twice with distilled water and stained with 0.4% SRB, then stored for 30 min at room temperature. Excess stain was removed from the plates by washing with 1% acetic acid twice. The plates were air-dried overnight. The protein-bound stain was solubilized with a 20 mM Tris-buffer solution for 15 min on an orbital shaker. The absorbance was measured with a microplate reader at a wavelength of 560 nm (Molecular Devices, Sunnyvale, CA, USA).

### 2.11. Cell-Migration Assay

CRC cells (HCT116 and DLD-1) were seeded into two-well (10^5^ cells in 100 µL media/well) well plates with a silicon insert in place and incubated for 24 h. After incubation, the medium was siphoned off and the insert was carefully removed, followed by the addition of fresh media (1.5 mL) containing LCC21 (1.25 µm). The wound was photographed under a microscope (Olympus CKX53 Cell Culture Microscope, Japan) immediately after treatment (0 h) and after 24 h, and wound closure was quantified with the aid of National Institutes of Health (NIH) ImageJ software (https://imagej.nih.gov/ij/ 12 July 2022).

### 2.12. Colony-Formation Assays

To assess the effects of LCC-21 treatment on the colony formation ability of colon cancer cell lines, we performed colony formation assay in accordance with the previously described protocol by Franken et al. [88]. In short, 400 cells were seeded in Corning^®^ 6 well plates (Sigma-Aldrich) and treated with LCC-21 (at the equivalence of 40% inhibitory concentration (IC_40_) values of HCT116 and DLD-1). Cells were allowed to grow for at least 1 week. Colonies were quantified using a Cell3iMager neo-scanner, and inhibitory effects of the drug on colonies as compared to control colonies were calculated in percentages (%).

### 2.13. Sodium Dodecylsulfate-Polyacrylamide Gel Electrophoresis (SDS-PAGE) and Immunoblot Analysis

Both the DLD-1 and HCT116 cell line groups treated with LCC-21 and the control group were harvested by trypsinazation. The total protein lysates of treated and untreated cells were then collected using a protein lysis buffer (RIPA buffer). Furthermore, 20 µg of total lysates were separated by SDS-PAGE using the Mini-Protean III system (Bio-Rad, New Taipei city, Taiwan) and transferred into polyvinylidene difluoride membranes using the Trans-Blot Turbo Transfer System (Bio-Rad) [88]. Membranes were then incubated with primary antibodies to react overnight at *−*4 °C. The following day membranes were incubated with secondary antibodies for 1 h. Proteins of interest were detected using enhanced chemiluminescence (ECL) detection kits (ECL kits; Amersham Life Science, California, CA, USA). Images were captured and analyzed by BioSpectrum^®^ Imaging System (Upland, CA, USA).

### 2.14. Tumor-Sphere Formation

Tumor spheres of HCT116 and DLD-1 cells were generated under serum-deprived culture conditions according to the method described by Dotse et al., 2016 [88]. In short, 2000 cells/well of colon cancer cells were seeded in six-well ultra-low-attachment plates (Corning, Corning, NY, USA), in serum-free media. Cells were allowed to aggregate and grow for 7 days. Those that were considered tumor spheres had a diameter of more than 50 μM and were characterized as dense, non-adherent spheroid-like masses. The spheres were counted using an inverted phase-contrast microscope.

### 2.15. Data Analysis

Statistical analyses were performed using Spearman and Pearson correlations to assess correlations between the target gene expressions in CRC. A Kaplan–Meier curve was used to present patient survival in CRC cohorts, with *p* < 0.05 accepted as being statistically significant.

## 3. Results

### 3.1. Identification of Differentially Expressed Genes (DEGs) in CRC

Differentially expressed genes obtained from CRC samples and adjacent noncancerous samples were extracted from the microarray datasets tallied from various studies. The results obtained from GSE81558 and GSE110223 datasets exhibited 51 and 26 CRC samples and normal samples, respectively. To further analyze using Venn diagrams, 110 overlapping upregulated genes from the two datasets were obtained (Figure 3A). The red and blue color in the volcano plot and heatmap diagram, respectively, show overexpressed and downregulated genes (Figure 3B–D). The volcano plot and heatmap revealed the statistical significance in the differential expression of tumor samples as compared to normal samples, denoted by –log10 (*p*-value) and –log (fold change), respectively.

### 3.2. LCC-21 Successfully Meets Required Drug-Likeness Criteria

The identification of potential drug candidates in the initial stages of drug discovery is dependent on the specific criteria based on the concept of drug-likeness [89]. Herein, we obtained all of the six physicochemical properties required for drug-likeness from SwissADME, which are represented on the bioavailability radar. Based on the results, LCC-21 successfully passed the minimum requirements of drug-likeness. These were based on the molecular weight (MW) of the compound (376.31 g/mol), lipophilicity (XLOGP3 = 4.31), polarity (TPSA = 76.00 Å^2^), solubility (ESOL = *−*5.34), saturation (fraction Csp3 = 0.11), and flexibility (number of rotations = 2) (Figure 4A). The standard criteria are as follows: molecular weight of a compound (Mw: ≤500 g/mol), flexibility (number of rotations: ≤10), solubility (log S (ESOL): ≤0–6), saturation (fraction Csp3: ≤1), polarity (TPSA: ≤140 Å^2^), and lipophilicity (XLOGP3: ≤0.7–5), all of which are recommended values. For further analysis, we applied the BOILED-egg prediction and estimated the BBB permeability of the compound. Based on the results, LCC-21 exhibited higher probability of BBB permeation, with the score of 0.02 (Figure 4B). Furthermore, we used a target prediction tool to identify all the target genes for LCC-21 (Table 1).

### 3.3. VEGFA/CTNNB1/MMP7/CD44 Oncogenic Signatures Are Overexpressed in CRC and Associated with a Poor Prognosis

Our computer-based analysis revealed that the messenger (m)RNA levels of VEGFA, CTNNB1, MMP7, and CD44 oncogenic signatures were upregulated in tumor samples compared to normal samples of patients with CRC tissues, with significant *p* values (<0.05) (Figure 5A–D). In further analyses, we found that the high expression levels of VEGFA/CTNNB1/MMP7/CD44 signatures were associated with significantly shorter survival times compared to those of patients with lower expression levels of these oncogenes (Figure 5E–H). Furthermore, we explored the TMN plot to compare VEGFA, CTNNB1, MMP7, and CD44 oncogenes in tumor and metastatic CRC samples from RNA sequencing data (RNAseq). Using the Kruskal–Walls test to compare data, we found that overexpression of VEGFA, CTNNB1, MMP7, and CD44 genes promoted primary tumor and cancer metastasis in CRC tissues (Figure 5I–L). We further investigated the correlations among VEGFA/CTNNB1/MMP7/CD44 signatures and found that, when all four oncogenes were combined for analysis, the predicted results showed positive correlations in the range of r = 0.16~0.27 of VEGFA with CTNNB1 and MMP7 in CRC patients (Figure 5M–O), with positive Spearman and Pearson correlation coefficients and *p* < 0.05 considered statistically significant.

### 3.4. Protein–Protein Interaction (PPI) Network Construction and Functional Enrichment Analysis

The high expression levels of *VEGFA/CTNNB1/MMP7/CD44* oncogenic signatures, which are associated with shorter survival times in CRC, suggest the predictive power of this signature. To explore further, we predicted the PPIs among these proteins. Based on the results, *VEGFA/CTNNB1/MMP7/CD44* oncogenes were found to be co-expressed and correlated within the same clustering network, and these correlation networks were based on curated data, co-expression, correlation, and experimentally validated data, which existed among all four of these oncogenes. (Figure 6A). Moreover, we analyzed the protein expression profiling by exploring network analyst, which is an online visual analytics platform. We selected the signaling network of KEGG pathway enrichment using the SIGnaling network platform. Interestingly, the enriched pathways displayed co-expression of *VEGFA/CTNNB1/MMP7/CD44* gene signatures within the same cluster when analyzed using the Igraph R package (Figure 6B). In addition, further analysis was performed on the enriched gene ontology (GO), including biological processes and biological pathways, with the criterial set to *p <* 0.05 (Figure 6C,D).

### 3.5. VEGFA/CTNNB1/MMP7/CD44 Genes Are Altered in CRC Tissues and Immune Cells

We applied the Oncoprint feature of cBioportal software, which revealed a volcano plot displaying overexpressed genes in CRC altered and unaltered cohorts, including (Figure 7A), we further explored the oncoprint to determine genetic mutations of the target oncogenes; *VEGFA/CTNNB1/MMP7/CD44*, which was based on percentages of separate genes due to amplification. Results of the analysis were as follows: *GSK3β* (2%), *MYC* (9%), *CTNNB1* (4%), *MMP7* (2.4%), CD*44* (2.2%), *KRAS* (9%), and *VEGFA* (2.4%) (Figure 7B). In a further analysis, we explored the tumor infiltrating immune cells tool to analyze and visualize the effects of gene mutations on immune cell infiltration in CRC. The analyzed results showed the mutation frequency of *VEGFA/CTNNB1/MMP7/CD44* genes displayed by the violin plots of immune infiltration distribution, including CD4^+^ T-cells, CD8^+^ T-cells, and macrophages in the wild-type compared to mutant tumors (Figure 7B–J).

### 3.6. VEGFA/CTNNB1CTNNB1/MMP7/CD44 Oncogene Expressions Are Correlated with Immune Cell Infiltration and Worse Prognosis in CRC

We further investigated the association between VEGFA/CTNNB1/MMP7/CD44 oncogenic expressions with selected immune cells in the tumor microenvironment (TME). Herein, we explored the correlation analysis between VEGFA, CTNNB1, MMP7, and CD44 oncogenes, and the immune infiltration cells (CD8+ T cells, CD4+ T cells, and macrophages). Interestingly, the results showed specific correlations of immune cell markers in colorectal adenocarcinoma (COAD), with lower infiltration levels CD8+ T cells, CD4+ cells, and high M2 macrophages (Figure 8A–D), with *p* < 0.05 considered significant.

### 3.7. VEGFA/CTNNB1/MMP7/CD44 Gene Expression Influence the Immune Landscape within the TME of Colorectal Cancer

We explored publicly available scRNA-seq datasets and single-cell RNA sequencing (scRNA-seq) datasets of CRC, which included the M-E-MAT-8410 and (9 patients; GSE144735) CRC tumors, and we also matched adjacent normal colon tissue for the purpose of generating a cellular map of CRC and their tumor microenvironment [81]. Within the CRC, a distinct population of malignant cells and multi-lineages revealed the presence of abundant immune cells within the tumor microenvironment (TME), as compared to stromal and malignant cells. Herein, we focused on cell types that were described as a putative population. Interestingly, we identified distinct populations of regulatory T cells, mast cells, macrophages, and myeloid cells; low expression levels of CD4^+^Tc and CD8^+^Tc cells; and exhausted CD8 T cells with the TME of CRC patients (Figure 9A). When observing the expressions of significantly regulated genes, which associates with the immune landscape of CRC based on similarities with the control, we found *VEGFA*, *CTNNB1*, *MMP7*, and *CD44* to be overexpressed by the majority in the malignant cells. For further analysis, merging expression patterns revealed distinct TME subpopulation with a high abundance of *VEGFA*, *CTNNB1*, *MMP7*, and *CD44* in regulatory T cells, mast cells, macrophages, myeloid cells; a low expression level of CD4^+^Tc and CD8^+^Tc cells; and exhausted CD8 T cells with the TME of CRC patients (Figure 9B–E).

### 3.8. In Silico Molecular Docking Showed Putative Binding of (NSC765689) LCC-21 with VEGFA/CTNNB1/MMP7/CD44

*In silico* molecular docking analyses revealed the potential inhibitory effects of (NSC765689) LCC-21 on oncogene markers, including *VEGFA/CTNNB1/MMP7/CD44.* Gibbs free energy results of protein–ligand interactions obtained through AutoDock Vina showed that LCC-21 displayed the highest respective binding energies of Δ = −8.1, −8.2, −9.0, and −8.0 kcal/mol, for *VEGFA*, *CTNNB1*, *MMP7*, and *CD44*, respectively (Figure 10A–D). For further analysis, we used Pymol and Discovery Studio to visualize the analytical results. The LCC-21/*VEGFA*, *CTNNB1*, *MMP7*, CD*44* interactions were stabilized by conventional hydrogen bonds, and their minimal distance constraints, van der Waals forces, carbon hydrogen bonds, and pi–pi interactions, with their respective amino acids, are shown in Figure 11A–D and Table 1.

### 3.9. Docking Analysis of VEGFA/CTNNB1/MMP7/CD44 with Their FDA Approved Inhibitors

For further analysis, we compared the docking analysis results of LCC-21 with standard inhibitors, sorafenib, SFRP-1, and batimastat for *VEGFA, CTNNB1*, and *MMP7*, respectively. Interestingly, the results showed that LCC-21 exhibited the highest binding affinities with *VEGFA*, *CTNNB1*, and MMP7 of (−(Δ = −8.1, −8.2, −9.0 kcal/mol), respectively, as compared to the sorafenib, SFRP-1, and batimastat for *VEGFA*, *CTNNB1*, and *MMP7* complexes, which displayed lower binding energies of (−(Δ = −7.3, −7.8, −8.6 kcal/mol), respectively (Figure 12A–C). Collectively, these structural simulations predicted LCC-21 to be a multi-target inhibitor with high confidence.

### 3.10. NSC765689 (LCC-21) Exhibited Anti-Proliferative and Cytotoxic Effects in NCI60 Human Colon Cancer Cell Lines

An initial single dose treatment of 10µM exhibited anti-proliferative effects on COLO205, HCC-2998, HCT-116, HCT-15, KM12, and SW620 cell lines and cytotoxic effects on HT29 (Figure 13A). Because the compound demonstrated potential anticancer activities on colon cancer cell lines at an initial dose of 10µM, its further compound was administered in vitro in a dose dependent manner to evaluate the (50%) growth inhibition (GI50) values, which ranged between 1.2 and 5.13 µM, with HCT-15, HCT116, KM12, SW620, HCC-2998, HT29, and COLO205 at 1.2 µM, 1.14 µM, 1.73 µM, 2.08 µM, 2.88 µM, 4.05 µM, and 5.13 µM. The tumor growth inhibition (TGI) values were also measured and ranged from 6.01 to 31.1 µM, with HCT-116, KM12, HCC-2998, HCT-15, HT29, COLO205, and SW620 at 6.01 µM, 7.13 µM, 7.9 µM, 10.7 µM, 15.1 µM, 17.9 µM, and 31.1 µM, respectively (Figure 13B, C). For further analysis, we identified increased expression levels of VEGF-A, CTNNB1, CD44, and MMP7 oncogenic signatures in different colon cancer cell lines (Figure 13D).

### 3.11. LCC-21 Decreases the Viability of CRC Cells through Modification of VEGFA/CTNNB1/MMP7/CD44 Oncogenic Signatures

To validate the above predictions, we evaluated the therapeutic activities of LCC-21 on the viability of CRC cell lines. The results showed that treatment with LCC-21 decreased the viability of HCT116 and DLD-1, with respective IC_50_ values of 3.2 and 2.8 µm (Figure 14A). Furthermore, LCC-21 also demonstrated inhibitory effects on DLD-1 and HCT116 migration and colony formation and effectively inhibited the sphere formation of these cells (Figure 14B–D). A Western blot analysis indicated that LCC-21 significantly decreased expression levels of VEGFA, CTNNB1, MMP7, and CD44. GAPDH was used as an internal control (Figure 14E).

## 4. Discussion

CRC remains a poorly characterized malignancy, and the third leading cause of cancer mortality rates globally [1]. Despite current treatment modalities, such as resection surgery, radiation, and chemotherapy, the overall survival rate is still less than 5 years for CRC [89,90]. The majority of CRC-related deaths are cases associated with cancer recurrence and metastasis; accordingly, most studies have shown that approximately 60% of CRC patients are expected to develop metastasis [91,92]. Therefore, in order to improve clinical outcomes, it is imperative to understand the mechanisms of CRC stemness and metastasis. Moreover, conventional chemotherapeutic strategies for CRC involve toxic drugs with severe side effects, which highlights the need to identify novel biomarkers for early diagnosis [93]. Additionally, combinations of molecular targeted agents have shown significant results as treatments for colorectal cancer; however, due to its heterogeneity, CRC patients eventually exhibit stemness and therapeutic resistance [94]. Therefore, exploring the mechanism underlying CRC therapeutic resistance is crucial for drug optimization [95].

Angiogenesis is one of the most common clinical phenotypes of CRC, and is crucial for tumor growth [40]. In addition, *VEGF* has been widely studied in relation to the development of novel drugs against CRC [96], and numerous reports have demonstrated high expression levels of *VEGFA* in CRC, which are associated with tumor angiogenesis, metastasis, and poor prognosis [33]. Therefore, understanding the regulatory mechanisms of *VEGFA* expression in CRC may lead to vital breakthroughs for novel therapies to fight various cancers [97]. In the current study, we utilized computational simulation, by exploring a bioinformatics analysis, and showed that VEGFA was elevated in CRC. Among the biomarkers evaluated in this study, these findings are in line with the above-mentioned study reports on the expression of *VEGFA* in CRC. Furthermore, recent studies revealed that chemokine receptors play vital roles in identifying the metastatic characteristics of tumor cells [98]. Multiple studies have shown that *VEGFA* positively regulates *Wnt/β-catenin* signaling through β-catenin, which is associated with angiogenesis and cancer stemness in CRC, thus suggesting the potential autocrine action of *VEGFA* [99,100,101].

Most anti *Wnt/β-catenin* therapies in CRC have being aimed at inhibiting cell invasion delay cancer progression, eliminate drug resistance, and preventing metastatic colorectal cancer (mCRC) [102,103,104], However, their contribution still remains unsatisfactory [105]. Accumulated studies have also shown that the *Wnt/β-catenin* signaling pathway interacts with angiogenesis markers in CRC [40], and this suggest its significance as a potential biomarker of angiogenesis. Herein, we further investigated the correlation between *VEGFA* and *β-catenin*. Our findings obtained from STRING and Genemania bioinformatics analyses confirmed the co-expression of *VEGFA* and *CTNNB1*, with the highest interactive confidence score of 0.900. Moreover, multiple studies have shown that *VEGFA* positively regulates *Wnt/β-catenin* signaling through *GSK-3β*, which is associated with angiogenesis and cancer stemness in CRC, thus suggesting the potential autocrine action of *VEGFA* [99,100,101]. Accordingly, we found that the *CD44* stemness marker is upregulated in CRC and also co-occurred with *VEGFA*, *STAT3*, and *CTNNB1* in CRC tissues. Additionally, others have shown that *VEGFA*-mediated development of CRC tumor angiogenesis was impeded by inhibition of *MMP7* [50,99,100]. *MMP7* is highly expressed in CRC epithelial cells and promotes metastasis [45,106]. In the current study, we found that *MMP7* was overexpressed in CRC tissues compared to adjacent normal tissues [46,107,108]. Previously, Fang et al. demonstrated the association of *VEGFA* and *MMP7* with immune invasion in CRC [109].

Our data also showed that increased levels of MMP7 were associated with *VEGFA* and were associated with tumor progression, metastasis, and immune invasion. These observations support our data, which demonstrated a marginal association between *MMP7* and *VEGF*, suggesting that *MMP7* could be one of the regulators of *VEGFA* activity in CRC cells. These findings thus suggest crosstalk among the *VEGFA/CTNNB1/MMP7/CD44* oncogenic signatures in regulating CRC progression, immune invasion, therapeutic resistance, metastasis, and poor prognosis. Immunotherapies have repeatedly been shown to impede the current cancer treatment landscape over the years, thus suggesting the need for development of novel immunotherapies. The identification of novel diagnostic markers for the development of these therapies requires molecular understanding of the TME, and its association with different prognostic markers [110,111]. ScRNAseq has been extensively utilized to study tumor heterogeneity and immune cells associated with the tumor in CRC [55,56]. Herein, we explored publicly available scRNA-seq datasets of CRC and found a high abundance of *VEGFA*, *CTNNB1*, *MMP7*, and *CD44* in regulatory T cells, mast cells, macrophages, and myeloid cells; a low expression level of CD4^+^Tc and CD8^+^Tc cells; and exhausted CD8 T cells with the TME of CRC patients, suggesting its associates with the immune landscape of CRC based on similarities with the control.

The collective analytical results using a bioinformatics analysis from TCGA groups of CRC tissues compared to adjacent normal tissues further confirmed that the overexpression of *VEGFA*, *CTNNB1*, *MMP7*, and *CD44* signatures were associated with angiogenesis, development of tumor growth, immune infiltration, resistance to therapies, and metastasis. To further investigate, we demonstrated the anticancer activities of LCC-21 as a target small molecule for the *VEGFA/CTNNB1/MMP7/CD44* signaling pathway in CRC. Our molecular docking analysis revealed the potential binding abilities of LCC-21 with *VEGFA*, *CTNNB1*, *MMP7*, and *CD44*, with results showing the highest respective binding energies of LCC-21: Δ = −8.1, −8.2, −9.0, and −8.0 kcal/mol, with the above-mentioned oncogenes. These results were higher than the docking analysis results of *VEGFA/CTNNB1/MMP7/CD44*, with their FDA approved inhibitors sorafenib, SFRP-1, and batimastat, which displayed lower binding energies of (−(Δ = −7.3, −7.8, −8.6 kcal/mol), respectively (Figure 12). Collectively, these structural simulations predicted LCC-21 to be a multi-target inhibitor with high confidence. We further validated these finding in vitro and demonstrated that LCC-21 inhibited colony formation, sphere formation, migration, and invasion, and these results were further confirmed by a Western blot analysis in HCT116 and DLD-1 cells. Thus, the inhibitory effect of LCC-21 on these angiogenic and onco-immunogenic signatures could be of translational relevance as potential CRC biomarkers for early diagnosis.

## 5. Conclusions

In conclusion, our results demonstrate that the overexpression of the *VEGFA/CTNNB1/MMP7/CD44* oncogenic signatures is associated with progression, immune infiltration, drug resistance, metastasis, and poor clinical outcomes in CRC. LCC-21, a novel multitarget small molecule, successfully suppressed *VEGFA*, *CTNNB1*, *MMP7*, and *CD44* signatures in CRC cell lines, thus suggesting that LCC-21 might be a potential novel candidate compound for inhibiting the *VEGFA/CTNNB1/MMP7/CD44* signaling pathway in CRC.

## Figures and Tables

**Figure 1 cells-12-00266-f001:**
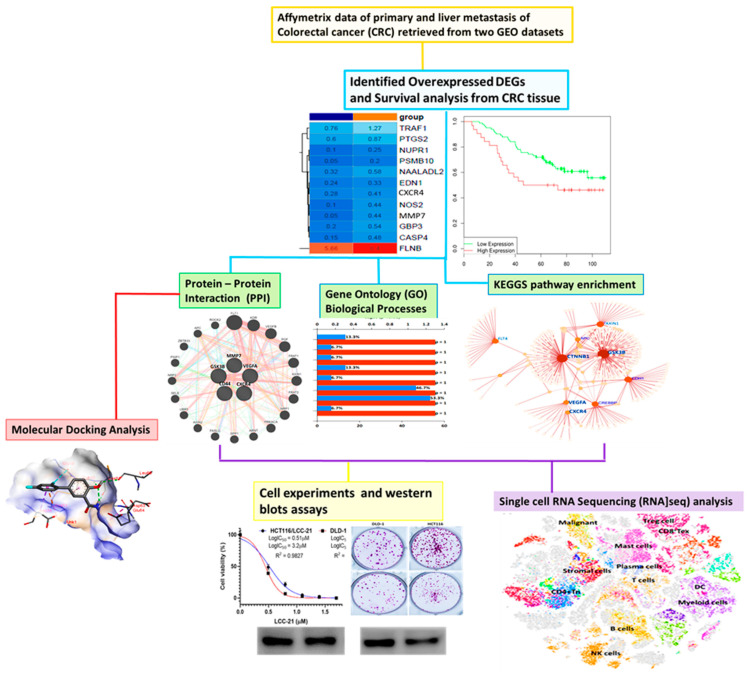
Schematic diagram representing the study design.

**Figure 2 cells-12-00266-f002:**
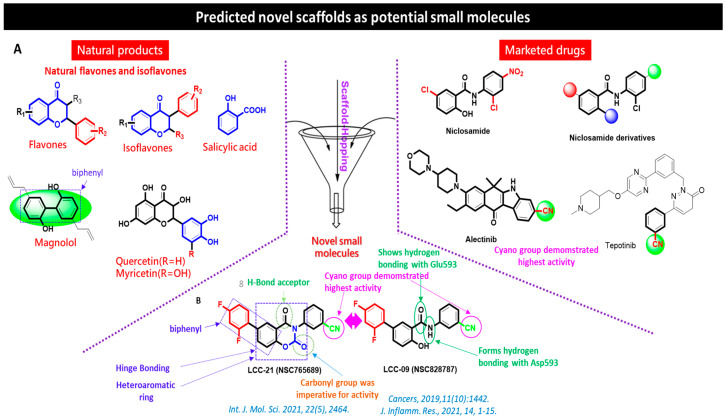
Chemical synthesis of the Novel Multi-Target small molecules NSC828787 (LCC09) and NSC765689 (LCC-21): (**A**) several novel series of 5-(2′,4, -difluorophenyl)-niclosamide derivatives based on difluorobiphenyl and niclosamide scaffolds, and (**B**) novel small molecule LCC-21 and LCC-09 consisting of magnolol, 2,4-difluorophenyl, and niclosamide functional fragments.

**Figure 3 cells-12-00266-f003:**
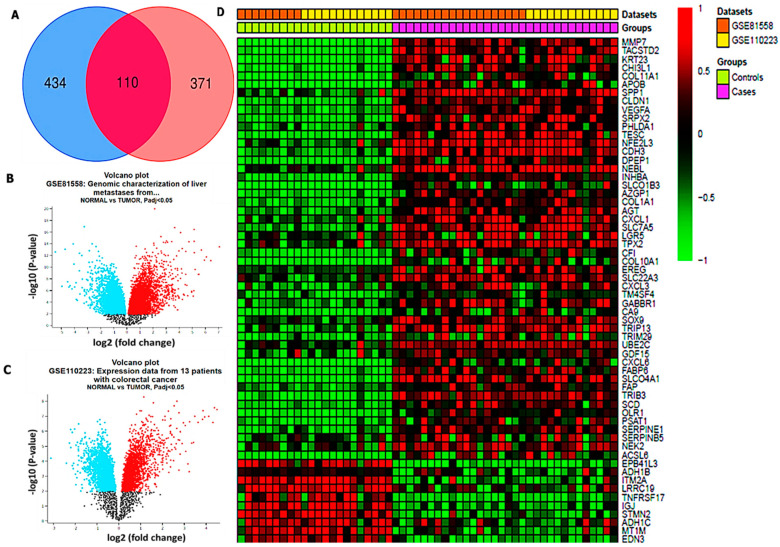
Microarray data mining of gene expression profiling in colorectal cancer (CRC). (**A**) Venn diagram showing overlapping genes obtained from the GSE81558 and GSE3110223 microarray datasets. (**B**,**C**) Volcano plots showing downregulated and upregulated genes in normal and tumor samples, respectively (at *p* < 0.05). (**D**) shows the heatmap of overexpressed overlapping genes in CRC.

**Figure 4 cells-12-00266-f004:**
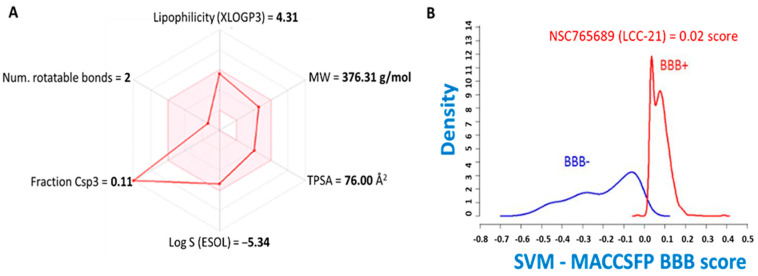
LCC-21 small molecules passed the minimum requirements for drug-likeness, medicinal chemistry, and ADME. (**A**) Physiochemical properties of LCC-21 represented on a bioavailability radar. (**B**) BBB permeability of LCC-21 (with 0.02 score). Table 1. Shows the specific protein targets of LCC-21.

**Figure 5 cells-12-00266-f005:**
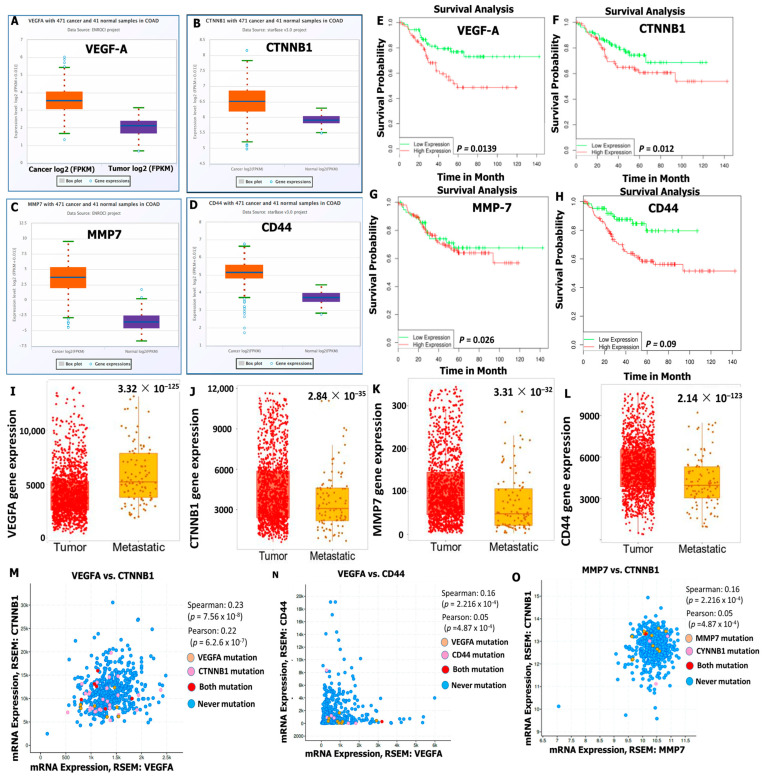
*VEGFA/CTNNB1/MMP7/CD44* oncogenic signatures are overexpressed in CRC and associated with a poor prognosis. Vascular endothelial growth factor-α (*VEGFA)*/beta catenin (*CTNNB1*)/matrix metalloproteinase (*MMP)-7*/cluster of differentiation 44 (*CD44*) oncogenic mRNA levels were overexpressed in colorectal cancer (CRC) tumor cohorts compared to normal samples with significant *p* values (**A**–**H**). Elevated mRNA levels of *VEGFA/CTNNB1/MMP7/CD44* were found to be associated with shorter survival times in CRC patients using the Wilcoxon test. (**I**–**L**) Upregulated gene expression of *VEGFA*, *CTNNB1*, *MMP7*, and *CD44* genes promoted primary tumor and cancer metastasis in CRC tissues. (**M**–**O**) The combination of all the four genes showed positive correlations in the range of r = 0.16~0.27 of *VEGFA* with *CTNNB1*, *VEGFA* with *CD44*, and *CTNNB1* with *MMP7* in CRC patients. Pearson correlation coefficients of *p* < 0.05 were considered statistically significant.

**Figure 6 cells-12-00266-f006:**
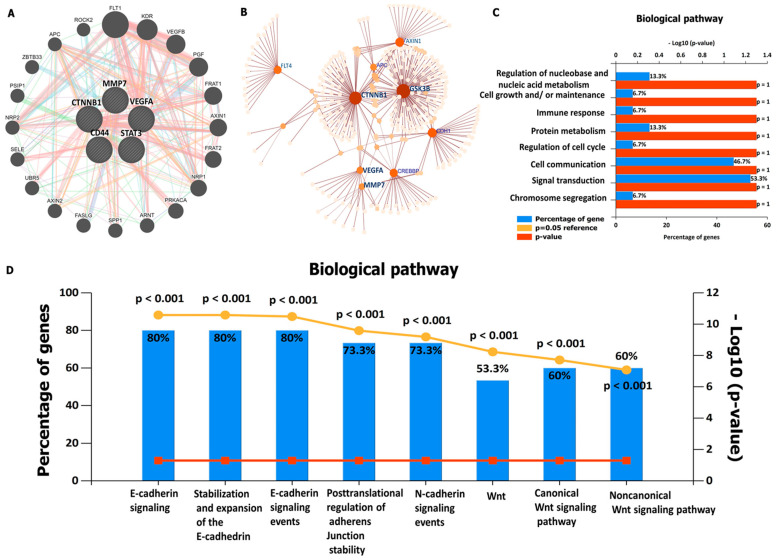
Correlation analysis network among Vascular endothelial growth factor-α (*VEGFA*)/beta catenin (*CTNNB1*)/matrix metalloproteinase (*MMP)-7*/cluster of differentiation 44 *(CD44)* oncogenic signatures. Protein–protein interactions (PPIs) are displayed after considering the gene neighborhood, gene co-occurrence, and co-expression of the clustering network. (**A**) Gene–gene interactions (GGI) between *VEGFA* with *CTNNB1*, *VEGFA* with *MMP7*, *VEGFA* with *CD44*, *CTNNB1* with *MMP7*, *CTNNB1* with *MMP7*, and *CD44* with *MMP7*. (**B**) KEGG pathway enrichment analysis from signaling network analysis. (**C**) Top 8 enriched biological processes. (**D**) Top 8 affected biological pathways, with *p*-value < 0.05 considered significant.

**Figure 7 cells-12-00266-f007:**
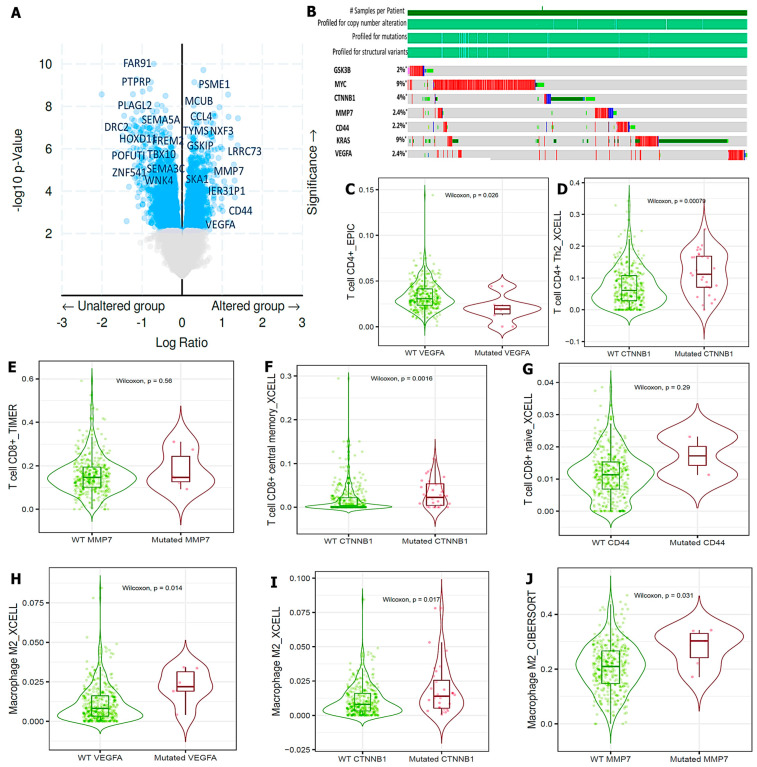
Vascular endothelial growth factor-α (*VEGFA*)/beta catenin (*CTNNB1*)/matrix metalloproteinase (*MMP)-7*/cluster of differentiation 44 *(CD44) oncogenes* were altered and amplified in colorectal cancer (CRC). (**A**) volcano plot showing upregulated genes in CRC altered and unaltered cohorts (**B**) Oncoprint analysis showed amplification (marked with *) of *VEGFA/CTNNB1/MMP7/CD44* based on percentages of separate genes, with *GSK3β* (2%), *MYC* (9%), *CTNNB1* (4%), *MMP7* (2.4%), *CD44* (2.2%), *KRAS* (9%), and *VEGFA* (2.4%) in CRC. (**C**–**J**) Mutation frequency of *VEGFA/CTNNB1/MMP7/CD44* Genes shown by the violin plots of immune infiltration distribution including CD4^+^ T-cells, CD8^+^ T-cells, and macrophages in the wild-type compared to mutant tumors.

**Figure 8 cells-12-00266-f008:**
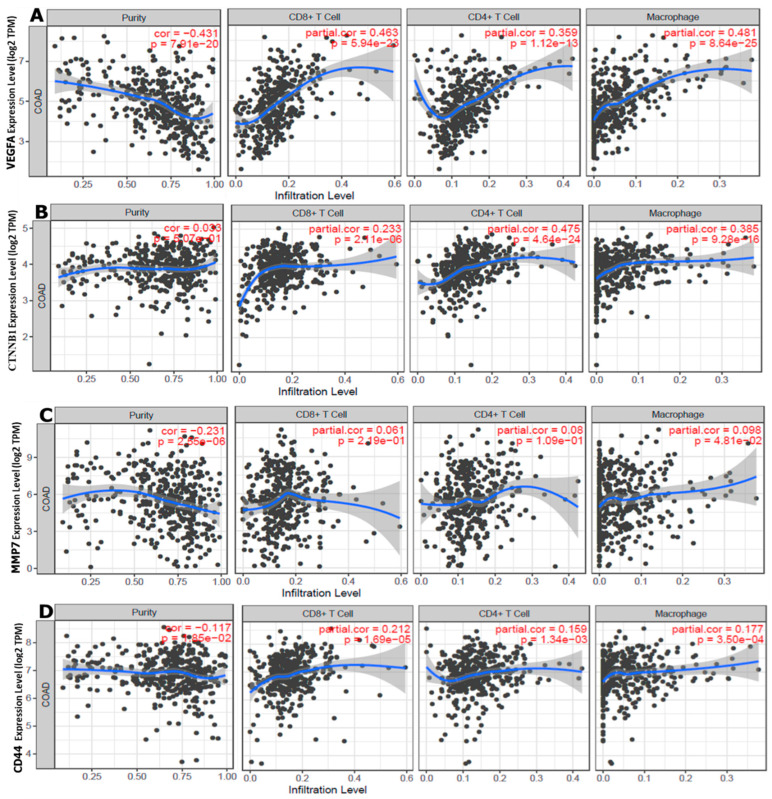
Correlation of vascular endothelial growth factor-α (*VEGFA*)/beta catenin (*CTNNB1*)/matrix metalloproteinase (*MMP7)*/cluster of differentiation 44 (*CD44*) expression with immune infiltrating cells in (COAD). (**A**) *VEGFA*, (**B**) *CTNNB1*, (**C**) *MMP7*, (**D**) *CD44*, and expression levels display associations with tumor purity and were positively correlated with lower infiltrating levels of CD8^+^ T cells and CD4^+^ cells, and a high infiltration level of M2 macrophages. The infiltration level was compared to the normal level using a two-sided Wilcoxon rank-sum test; *p* values of <0.05 were deemed significant.

**Figure 9 cells-12-00266-f009:**
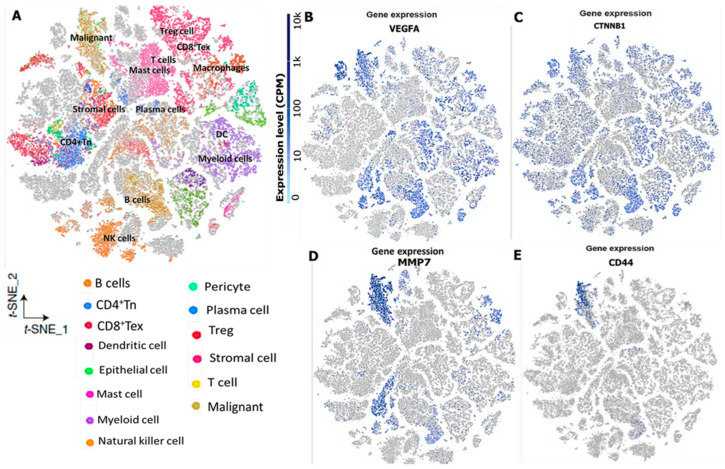
Single cell sequencing profiling of *VEGFA/CTNNB1/MMP7/CD44* in the tumor microenvironment of primary CRC. (**A**) T-distributed stochastic neighbor embedding (t-SNE) plot of single-cell RNA-seq data from M-E-MAT-8410 and (9 patients; GSE144735) CRC tumors. Proportions of the immune cell types in CRC tissue and normal colon tissue on average (left). (**B**–**E**) The expression distribution of *VEGFA/CTNNB1/MMP7/CD44* in different cell types, including regulatory T cells, mast cells, macrophages, myeloid cells; low expression level of CD4^+^Tc and CD8^+^Tc cells; and exhausted CD8 T cells with the TME of CRC patients.

**Figure 10 cells-12-00266-f010:**
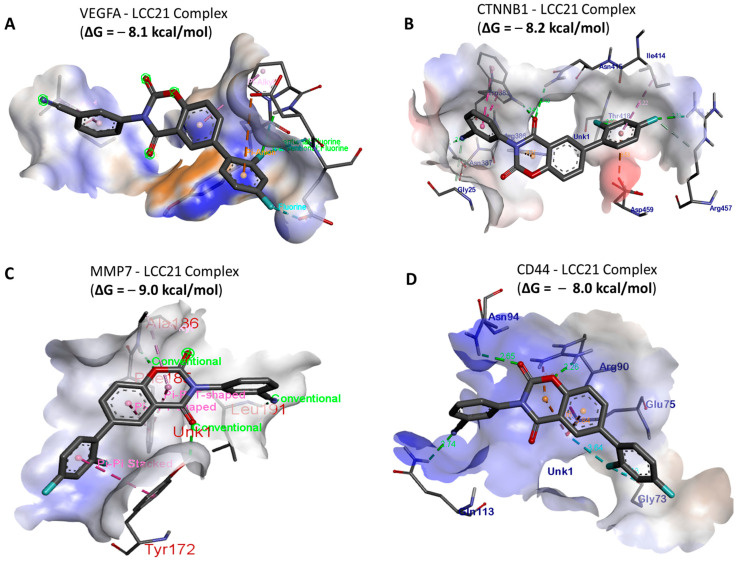
LCC-21 was shown to be a potential small-molecule inhibitor of multi-oncogenic proteins. (**A**–**D**) 3D structures of Vascular endothelial growth factor-α (*VEGFA*)/beta catenin (*CTNNB1*)/matrix metalloproteinase (*MMP)-7*/cluster of differentiation 44 (*CD44*) oncogenes-interactions (left) in complex with LCC-21, with the highest respective Gibbs free binding energies of (−(Δ = −8.1, −8.2, −9.0, and −8.0 kcal/mol).

**Figure 11 cells-12-00266-f011:**
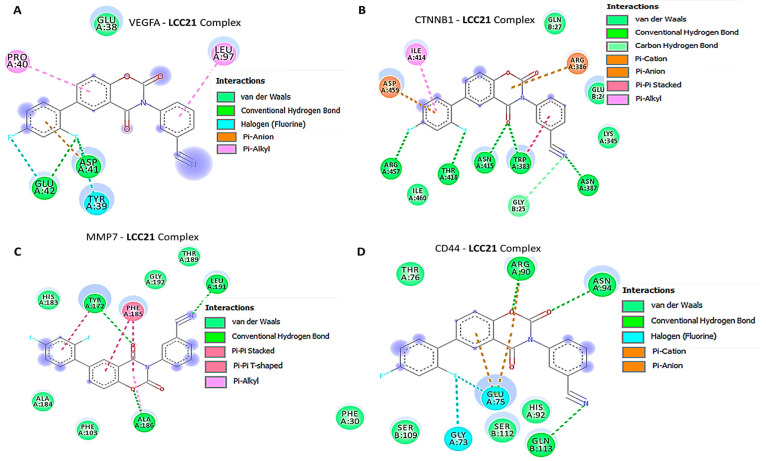
Two-dimensional (2D) structural analysis using Discovery Studio showing (**A**–**D**) ligand–receptor interactions between the atoms involved in H-bonding, amino acids and bond distances, van der Waals forces, carbon hydrogen bonds, and pi–pi interactions, with their respective amino acid of *VEGFA/CTNNB1/MMP7/CD44*-LCC-21 complexes using Discovery Studio (right). Table 2 summary of ligand–receptor interactions with their respective amino acids.

**Figure 12 cells-12-00266-f012:**
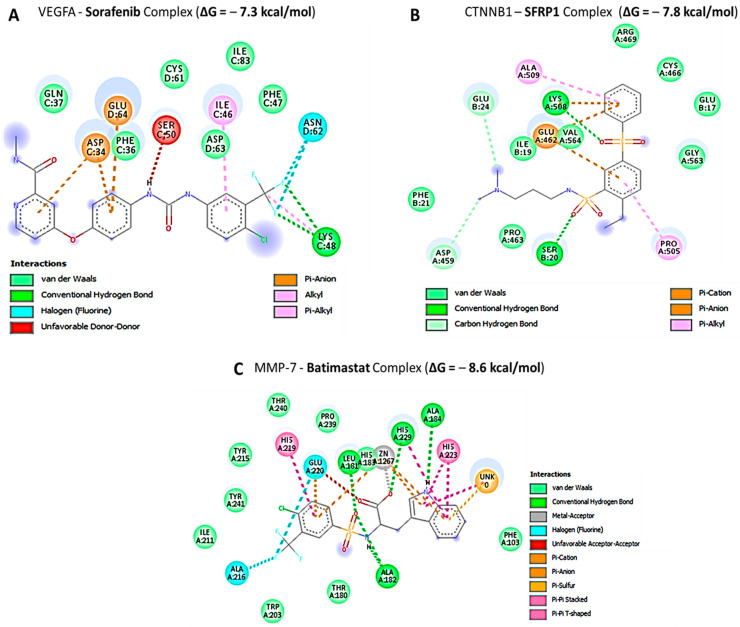
Two-dimensional (2D) structural analysis of sorafenib, SFRP-1, and batimastat in complex with *VEGFA*, *CTNNB1*, and *MMP7* (**A**–**C**), respectively.

**Figure 13 cells-12-00266-f013:**
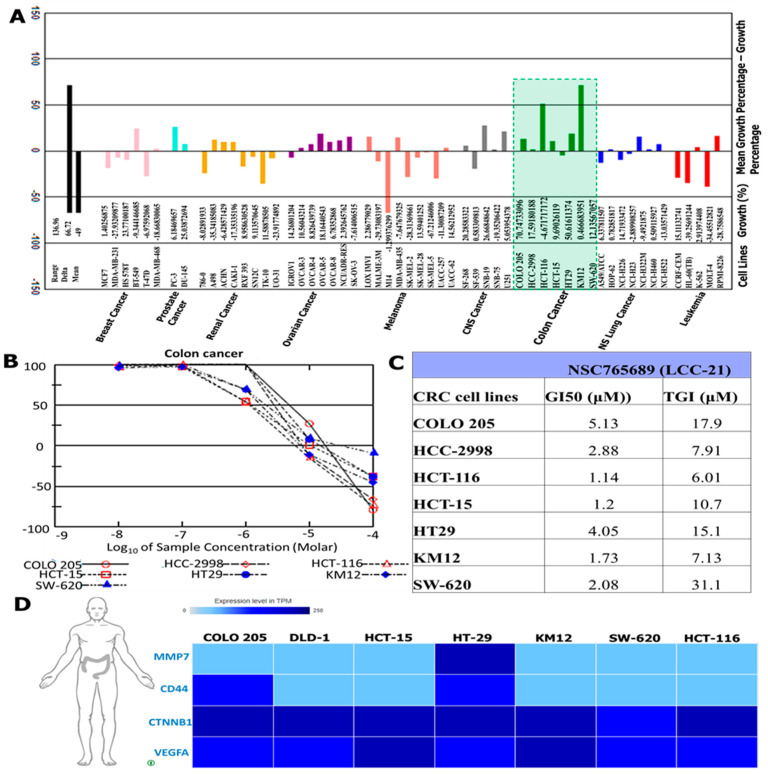
NSC765689 (LCC-21) demonstrates anticancer activities on colon cancer cell lines. (**A**) An initial 10 µM single-dose treatment displayed anti-cancer effects of LCC-21 on colon cancer cell lines. (**B**,**C**) LCC-21 was able to effect 50% growth inhibition (GI50) and tumor growth inhibition (TGI) at dose-dependent treatment. (**D**) Overexpression of VEGF-A, CTNNB1, CD44, and MMP7 genes in different colon cancer cell lines.

**Figure 14 cells-12-00266-f014:**
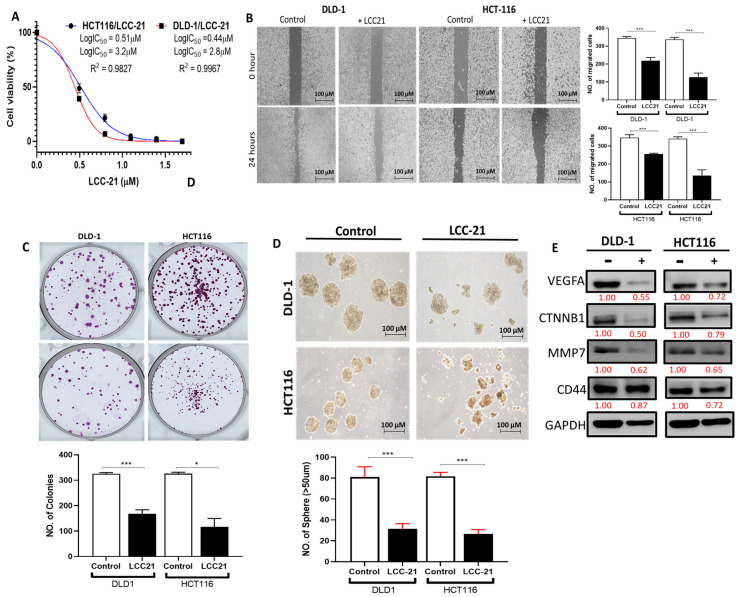
LCC-21 treatment suppressed the viability of colorectal cancer (CRC). (**A**) A cell viability assay demonstrated that LCC-21 effectively decreased the cell viability of CRC cell lines in dose-dependent and time-dependent manners. The experiments were performed three times, each with three replicates at each concentration The 50% inhibitory concentration (IC_50_) values in the two cell lines (HCT116 and DLD-1) are indicated. (**B**–**D**) LCC-21 inhibited migration, colony, and sphere formation by DLD-1 and HCT116 cells. Images of the colony and spheroids are shown in the left upper panel, with quantification of the results in the right panel. (**E**) Western blot analysis shows that the LCC-21 significantly suppressed expression levels of vascular endothelial growth factor-α (*VEGFA*)/beta catenin, (*CTNNB1*)/matrix metalloproteinase, (*MMP)-7*/cluster of differentiation 44 (*CD44*), compared to the untreated group. GAPDH was used as an internal control. * *p* < 0.05, *** *p* < 0.01.

**Table 1 cells-12-00266-t001:** Common names, Uniprot and ChEMBL identifications (IDs), and target classes of specific protein targets of LCC-21.

Target	Common Name	Uniprot ID	ChEMBL ID	Target Class
Cyclin-dependent kinase 9	CDK9	P50750	CHEMBL3116	Kinase
Tyrosine-protein kinase JAK3	JAK3	P52333	CHEMBL2148	Kinase
Vascular endothelial growth factor-α	VEGFA	P00533	CHEMBL203	Kinase
Glycogen synthase kinase-3β	GSK3β	P49841	CHEMBL262	Kinase
Mitogen-activated protein kinase	MAPK14	Q16539	CHEMBL260	Kinase
Signal transducer and activator of transcription 3	STAT3	P40763	CHEMBL4026	Transcription factor
Cyclin-dependent kinase 1 Catenin β1	CDK1 CTNNB1	P06493 P36861	CHEMBL308 CHEMBL309	Kinase Kinase
MYC proto-oncogene	MYC	P35557	CHEMBL3820	Enzyme
Matrix metalloproteinase 7	MMP7	P14780	CHEMBL321	Enzyme

**Table 2 cells-12-00266-t002:** Analytical summary table showing interactions of LCC-21 with *VEGFA/CTNNB1/MMP7/CD44*.

VEGFA-LCC-21 Complex	(Δ = −8.1 kcal/mol)	CTNNB1-LCC-21 Complex	(Δ = −8.2 kcal/mol)
*Type of interactions and number of bonds*	*distance of interacting A*min*o acids*	*Type of interactions and number of bonds*	*distance of interacting A*min*o acids*
*Conventional Hydrogen bond (3)*	LEU (2.48 Å), CYS61 (2.04 Å), ASP63 (2.39 Å)	Conventional Hydrogen bond (5)	ARG457 (1.17 Å), THR418 (2.12 Å), ASN415 (1.87 Å), TRP383 ASN387 (3.17 Å) TRP383 (2.17 Å)
Van der Waals Forces	VAL33, LEU33, SER50, GLU64, ILE48, LYS48, ASN62	Van der Waals Forces	ILE460, GLU24, LYS348
Halogen	GLY59	Carbon hydrogen bond	GLY25
Pi–pi anion	ASP34	Pi-cation	ASP387
Amide pi-stacked	ASP63	Pi-anion	ARG386
Pi-Alkyl	CYS51, CYS60		
**MMP7-LCC-21 Complex**	**(Δ = −9.0 kcal/mol)**	**CD44-LCC-21 Complex**	**(Δ = −8.0 kcal/mol)**
*Type of interactions and number of bonds*	*distance of interacting A*min*o acids*	*Type of interactions and number of bonds*	*distance of interacting A*min*o acids*
Conventional Hydrogen bond (3)	LEU19 (2.24 Å),TRY172 (1.29 Å), ALA186 (2.00 Å)	Conventional Hydrogen bond (3)	ARG90 (3.64 Å), ASN94 (2.39 Å), GLN113 (3.00 Å)
Van der Waals Forces	HIS183, ALA184, GLY192, GLY190, THR189	Van der Waals Forces	CYS77, THR76, PHE30, SER109, HIS92, SER112
Pi–pi cation	PHE185	Halogens	GLU75, GLY73

## Data Availability

Links for data availability are provided in the materials and methods section.

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
