# Peer review of "Preclinical Evaluation of a Novel Small Molecule LCC-21 to Suppress Colorectal Cancer Malignancy by Inhibiting Angiogenic and Metastatic Signatures"

_cells, 2023, doi:10.3390/cells12020266_

Round 1

Reviewer 1 Report

In this study, Mokgautsi et al. identified a new multi-target small molecule LCC-21 through bioinformatics analysis, which can inhibit VEGFA/CTNNB1/MMP7/CD44 in CRC cells, and confirmed these findings in vitro. Overall, this paper could be an interesting contribution to Cells. However, I think there are still some shortcomings in the work:

1.     The P value in Figure 5G is not clear, so no definitive conclusion can be drawn that high expression of MMP7 is associated with shorter survival times of CRC patients.

2.     The P value in Figure 5H is 0.09, so no definitive conclusion can be drawn that high expression of CD44 is associated with shorter survival times of CRC patients.

3.     Spearman and Pearson correlation coefficient |r| is less than 0.3 in Figure 5M-O. Therefore, the authors suggested that the correlation between VEGFA/CTNNB1/MMP7/CD44 is not convincing.

4.     In the Results section, the authors wrote VEGFA/CXCR4/GSK3β/MMP7 oncogene expressions are correlated with immune cell infiltration and worse prognosis in CRC, but the correlation between VEGFA/CTNNB1/MMP7 and immune cell infiltration was analyzed in the corresponding Figure section. In addition, the Figure 8 legend shows the correlation analysis between VEGFA/CTNNB1/MMP7/CD44 and CD8+T cells, CD4+T cells and M2 macrophages, but does not show the correlation analysis content of CD44. And | r | is less than 0.3 and P value is more than 0.05.

Minor revision

1.    The content in the Figures and legends should be consistent with the Results section.

2.    Original images of immunoblots with no cuts should be added to the supplementary material.

Reviewer 2 Report

 In the article titled “Preclinical Evaluation of a Novel Small Molecule LCC-21 to Suppress Colorectal Cancer (CRC) Malignancy by Inhibiting Angiogenic and Metastatic Oncogenic Signatures” Mokgautsi et al. characterized the biological activity of LCC-21, a new small molecule, in different colorectal cancer cell lines.

The manuscript is clear and scientific data in the draft are  presented in a well-structured manner.

On the contrary, I am of the opinion that the raw data are not scientifically evaluable, due to an evident unreliability of blot pictures.

From my point of view, it would be appropriate that the Authors perform new biological experiments for characterization of LCC-21 activity in normal colon cell lines, in order to compare this results with  standard therapeutic regimens for CRC,  sorafenib and  bevacizumab.
